# Cost-effectiveness of the Healthy Options group psychosocial intervention for perinatal women living with HIV and depression in Tanzania

**Happiness Pius Saronga**[1]*, **Sylvia Kaaya**[2], **Mary C. Smith Fawzi**[3]

**1** Department of Behavioural Sciences, Muhimbili University of Health and Allied Sciences, Dar es Salaam, Tanzania, **2** Department of Psychiatry and Mental Health, Muhimbili University of Health and Allied Sciences, Dar es Salaam, Tanzania, **3** Department of Global Health and Social Medicine, Harvard Medical School, Boston, Massachusetts, United States of America

* sarongahappiness@yahoo.com

**Data Availability Statement:** The dataset supporting the conclusions of this article is included within the article and supporting file.

## Abstract

Healthy Options is a psychosocial support group intervention facilitated by community-based health workers (CBHWs) to reduce symptoms of depression in perinatal women living with HIV in Tanzania. The objective of this study was to determine incremental cost-effectiveness of Healthy Options intervention in comparison to enhanced usual care for depression (EUDC) intervention. This study is a cost-effectiveness analysis of Healthy Options intervention. The primary outcome for the Healthy Options intervention was level of depressive symptoms. We estimated disability adjusted life years (DALYs) by considering life years lived with disability and years of life lost due to premature mortality resulting from depression. This study applied ingredients approach to cost all resources used in the intervention. We estimated total cost, unit cost, and incremental cost-effectiveness ratio (ICER) from a health care provider perspective. We used 3-year time horizon, univariate sensitivity analysis, and adjusted costs to 2017 value. Healthy Options intervention demonstrated effectiveness in reducing depressive symptoms among pregnant women with HIV in Tanzania. Total cost of Healthy Options was $319,729. Cost per woman treated was $883. ICER at 6 weeks postpartum is $89,699 per mean decrease in depression score and $310,030 per mean decrease in depression score at 9 months. ICER per DALY averted is $24,754 at 6 months and $4,169 at 9 months. Benefits of Healthy Options are sustained through 9 months postpartum. Healthy Options is nevertheless not cost-effective because ICER is above cost-effectiveness threshold. However, since mental health care is scarce in Tanzania, working with CBHWs is likely to offer effective intervention for maternal depression among women with HIV and it can be a less costly alternative to formal mental health professionals.

## Background

Depression contributes significantly to the global burden of disease and is a leading cause of disability worldwide [1]. The burden of depression for the affected individuals, their families,

**Funding:** The National Institute of Mental Health, R01-MH100338, funded this study. SK and MCSF received the award. The funder had no role in study design, data collection and analysis, decision to publish, or preparation of the manuscript.

**Competing interests:** The authors have declared that no competing interests exist.

communities as well as the economy is substantial [2, 3]. The cost of depression involves health care utilization in the form of direct treatment costs as well as indirect economic costs in form of lost productivity from premature mortality and morbidity, absenteeism, and impairment [3, 4]. Other consequences of depression may include suffering, reduced quality of life, cost of informal care and long-term effects such as reduced economic security, among others [5, 6].

In addition, depression is common among people living with HIV and even more so among HIV-positive pregnant women [7, 8]. Depression during pregnancy has been linked to a complex interaction of factors, such as intimate partner violence, lack of social support, poor socioeconomic status, and food insecurity, among others [9–11]. Depression is very common during the perinatal (antenatal and postnatal) period, and antenatal depression highly determines presence of postnatal depression. Negative effects of perinatal depression may apply to the woman as well as her child. Perinatal depression may lead to reduced quality of life for the woman, poor maternal nutrition, birth complications, poor child nutrition, and stunted child growth with a negative long-term impact on the child's health, cognitive development, and academic achievement. This can lead to long-term effects on their economic security with a broader impact on the overall economy as well [2, 3, 12].

In Tanzania, depression among pregnant women is common. For example, a study conducted in a peri-urban area of Tanzania, reported a prevalence of 40% [9], while another study conducted among women attending antenatal clinic (ANC) at a national hospital located in an urban area reported a prevalence as high as 78% [10]. With this level of burden, screening for depression in ANC has been proposed [9, 10]; however, in practice depression is not routinely screened for during ANC [13]. This presents a missed opportunity to treat this common condition affecting many women living with HIV at a most critical moment when negative effects of lack of access to care befall not only the woman but her children as well [5]. Untreated depression may lead to negative coping mechanisms such as drug and alcohol misuse, as well as limit self-care, e.g. poor nutrition and reduced health seeking behaviour [14]. All of these factors can negatively affect health and quality of life for both the mother and her child [15]. Furthermore, depression has a negative impact on ART adherence, HIV prognosis and survival among women living with HIV [7, 13].

To improve health outcomes and quality of life in perinatal women living with HIV and depression, a psychosocial support group intervention (named Healthy Options) facilitated by lay community-based health workers (CBHWs) was piloted in Dar es Salaam, Tanzania [16]. The intervention incorporated problem-solving therapy (PST) and cognitive behavioural therapy (CBT). This was a first intervention of its kind in Tanzania and results from the trial indicated that Healthy Options is effective in treating depression in perinatal women living with HIV [17]. This study was conducted to assess cost-effectiveness of the Healthy Options intervention in treating depression in perinatal women living with HIV in comparison to enhanced usual care for depression (EUDC). Given limited resources and the need to maximize health, economic analysis information is useful to show the value of treating depression in perinatal women living with HIV in Tanzania. Current evidence indicates cost-effectiveness information of the treatment for perinatal depression is lacking for low- and middle-income countries (LMICs) [18], thus this study will contribute to addressing this information gap.

## Methods

### Study design and setting

The aim of this study was to estimate incremental cost-effectiveness of Healthy Options intervention in managing perinatal depression in HIV positive women. Cost-effectiveness analysis was conducted from a health care provider perspective. Healthy Options was a group

psychosocial intervention for perinatal women living with HIV and depression in Dar es Salaam Tanzania. Detailed methodology used in the Healthy Options intervention is published elsewhere [16, 17]. Nonetheless, a brief description and context of the intervention is provided in the following section.

## Healthy Options intervention study

Healthy Options was a pair-matched cluster-randomized controlled trial, conducted in government health facilities with reproductive and child health services offering prevention of mother to child transmission of HIV (PMTCT) services. The study randomly selected 16 clusters, which were health facilities including health centres and their satellite dispensaries in three districts of the Dar es Salaam region in Tanzania. From the study clusters, women were eligible for enrolment into the trial if they were 18-years-old and above, were pregnant with gestational age of no more than 30 weeks, were receiving ART, had symptom severity comparable to major depression as assessed by the Patient Health Questionnaire-9 (PHQ-9), had no symptoms suggestive of moderate to high suicide risk, given their depressive symptoms, and planned to access postnatal care at the study sites. HIV-positive pregnant women were screened for eligibility and invited to participate in the study.

Randomization into intervention or control groups was done by clusters with a 1:1 allocation ratio. Clusters were pair-matched by geographic location within study districts (Ilala, Kinondoni, and Temeke). Within each pair match we used a random number generator to determine which cluster would be the intervention cluster. Clusters were stratified by district and pair-matched in each district using census classification of facility location as urban or peri-urban. Women enrolled in the intervention and control facilities were matched on socio-demographic characteristics age, marital status, education level, employment status, household economic indicators; as well primary and secondary outcomes for the randomized controlled trial at baseline- depression, intimate partner violence, social support scale, self-efficacy scale, hope, HIV-related stigma scale, and ART adherence. The average cluster size was 46 people (range 10–98). Facilities, patients, outcome assessment, and analysis were not blinded.

Both arms received EUDC training. The EUDC training was designed to improve the capacity of all the facilities to provide screening, treatment, and referral for women presenting with elevated levels of depressive symptoms. Clinical staff were trained with the expectation that they will use the knowledge to screen and treat depression as there was no routine screening of depression at that time, caused by limited number of mental health professionals in Tanzania. Antenatal care services in Tanzania are typically offered by clinical staff who have limited skills for assessment of depression and intervention among women with major depression.

EUDC training was conducted for one day and included medical officers, clinical officers, nurses, medical social workers, and medical attendants. The training was based on the World Health Organization (WHO) mhGAP guidelines, which focuses on screening for depression; utilizing an algorithm for diagnosis assessment and management; attending to aspects of depression that can be addressed in a primary care setting, including exploration of current stressors, psychoeducation, reactivating social networks, offering structured physical activity, and follow-up; and identifying patients that need referral for specialized psychiatric care [19].

Intervention sites then received Healthy Options on top of the EUDC training. Healthy Options includes prenatal group sessions of problem-solving therapy (PST) and cognitive behavioral therapy (CBT) sessions for individuals showing continued depressive symptoms at six weeks post-delivery. PST and CBT are commonly used therapeutic approaches. The structured psychological treatment was provided in a group format by trained CBHWs. PST and

CBT were provided at two different time points. The PST groups were delivered before delivery and included one orientation and six psychological group therapy sessions with 15–20 women. The PST delivery processes promoted awareness of one's own strengths and assertive communication skills and included acknowledging problems and identifying strategies for addressing them, focusing on positive thinking approaches, including using dreaming to develop action plans and foster hope, as well as skills for assertiveness and goal setting.

Depressive symptoms were assessed six weeks post-delivery using the PHQ-9 scale and women who scored a 9 or above indicating clinically significant depression were invited to participate in eight CBT group sessions adapted from the WHO Thinking Healthy Intervention [20]. CBT was delivered in small groups of 8–10 women and included discussing the link between one's thoughts and feelings; sharing practical skills of caring for their infants and addressing daily challenges; discussing strategies for managing stress; fostering support from their social networks; and offering feedback on their CBT homework assignments with the goal of addressing depressed mood through cognitive restructuring. The group sessions were all facilitated in Kiswahili, the local language and lasted approximately 2–3 hours per session. CBT involved main sessions and smaller break-out groups for discussion and skills building. PST group sessions were held in a variety of venues, such as health facilities, schools, and religious sites close to health facilities. CBT sessions were delivered at 8 reproductive and child health (RCH) clinics in total. The interventions did not mask facilities nor participants because of their nature. CBT was provided to 46 women with elevated depressive symptoms at six weeks follow-up after childbirth.

CBHWs providing Healthy Options had undergone a two-week in-class psychosocial training, followed by once weekly skills practice sessions for four weeks. All CBHWs had prior experience as ancillary workers providing HIV-related peer psychosocial support group counselling sessions hence placing them at a very good place to provide the intervention.

## Intervention outcomes

The primary outcome measured from the Healthy Options trial was level of depressive symptoms. Depressive symptoms were measured by the PHQ-9, which is scored on a scale from 0 to 27 with a locally validated cut off point at 9 or above to be consistent with major depressive disorder [21]. Outcome was assessed at baseline, 6 weeks postpartum as well as 9 months postpartum. Intention-to-treat analysis was performed to estimate intervention outcomes using generalized estimating equations accounting for clustering. All models included a fixed effect for district of the health facility to account for stratification by district prior to randomization. As a sensitivity analysis to assess the potential for baseline imbalance to affect the results, we conducted multivariable regression adjusting for hypothesized confounders and variables that showed imbalance at baseline, including age, marital status (married or living with partner versus other), completed secondary education, and employment (formal or self-employment). We also assessed potential effect modification of the effect of the intervention on the primary outcome of depression at 9 months postpartum by baseline values of the depression score, disclosure of HIV status to partner, report of intimate partner violence, social support, hope, and HIV-related stigma.

Level of depressive symptoms is an intermediate outcome to the intervention; it was subsequently transformed into Disability-adjusted life years (DALYs). A DALY is a measure of life years lived with disability (YLD) and years of life lost due to premature mortality (YLL) resulting from depression. YLD is a function of disability weight, number of depression cases and duration of illness. YLL is a function of number of deaths and standard life expectancy at age of death. In the calculation of YLD, we considered disability weight for moderate depression episodes[22] as the participants were treated outpatient, we assumed severe depression would

require hospitalization. On the duration of depression, the assumption is that a case of depression would last on average for 6 months. In YLL calculation, we considered 30 as the average age at death, 38 as the standard life expectancy at age of death, this is because life expectancy at birth for females in Tanzania was 68 years in 2017 [23].

## Costing of the intervention

This study applied ingredients approach to cost all activities and resources used in the Healthy Options intervention [24]. Timeline for the analysis was 3 years from April 2014 to March 2017, during which intervention activities occurred. Research costs were excluded from the analysis to focus only on incremental cost of the intervention. Research costs are costs that were related to ethical clearance and randomization. These costs were excluded because the focus was to include only costs that would be incurred if the intervention was to be adopted and scaled up in a non-research setting.

Intervention inputs composed of office consumables cost that excluded software expenses (NVIVO, STATA, endnote) and comprised 75% of cost of two computers and their unlimited power supplies (UPS)- the rest 25% of cost were used in non-intervention activities hence not counted into the study cost; communication consisted 75% of all communications costs incurred- the rest 25% of cost were used in non-intervention activities hence not counted into the study cost; personnel costs included 1 psychiatrist, facilitators (lay CBHCWs- we started with 22 part-time facilitators in 2015 then reduced to half the number in 2017), 5 part-time supervisors and 1 full-time program coordinator; printing cost for training materials (training sessions of lay CBHWs and health providers in the enhanced standard of mental health care); transport reimbursement for training participants (CBHWs and health providers); refreshments for training participants (CBHWs and health providers) and psychosocial support group sessions; travel reimbursement and stipend for participants attending group sessions (Tanzanian Shillings 12,500 per woman per session- equivalent to $5.5).

Information for resources used in the intervention was retrieved from project records. Additionally, we used existing market prices to calculate the cost of buildings (assuming space requirement of 64 square meters at each intervention health facility) and the cost of furniture (assuming requirement for 1 table, 3 chairs, and 4 benches). Buildings and furniture were annuitized using straight-line methodology and useful life of 30 and 5 years respectively. Details on resources and cost are provided in S1 Table.

For cost comparability across time, we adjusted past year costs to 2017 equivalent costs using GDP deflator, the measure of general price level in the economy [25]. We obtained GDP deflator values from the World Bank website [26]. Since cost data were of past years and were reported in Tanzanian Shillings, we deflated the costs to 2017 and converted the adjusted costs to United States Dollars ($) using the 2017 exchange rate as indicated by the Bank of Tanzania [27]. We calculated total cost by adding up all cost categories, and unit cost as cost per participant, by dividing total cost by the total number of participants. We report inflation-adjusted intervention cost.

## Cost-effectiveness analysis of the intervention

To determine the cost-effectiveness of the Healthy Options intervention, we calculated incremental cost-effectiveness ratio (ICER). We calculated ICER as the difference in cost divided by the difference in depression score and DALYs averted between the Healthy Options intervention and the EUCD. The ICER per DALY averted for Healthy Options was compared to existing cost-effectiveness thresholds suggested by the World Health Organisation (WHO) and Ochalek et al. (2018) [28, 29].

### Sensitivity analysis

To check robustness of estimates we performed one-way sensitivity analysis. We varied input assumptions that may most likely affect cost-effectiveness results. We specifically varied disability weight used in DALY calculation (we used weights for mild and severe depression [22] as lower and upper bound values) and intervention cost (we varied total cost by +-20%) consistent with a similar study [30]. We used Microsoft Excel to perform economic analysis.

### Ethics approval and consent to participate

Ethical review boards at Harvard Medical School in the U.S. and the National Institute for Medical Research in Tanzania approved the study. All participants in the Healthy Options intervention provided written informed consent prior to participating in the study. All methods were performed in accordance with the Declaration of Helsinki. The Healthy Options trial is registered at Clinicaltrials.gov (NCT02039973).

## Results

### Healthy Options intervention outcomes

A total of 742 women were enrolled in the trial. Baseline socio-demographic characteristics and health outcomes of the two arms were similar. 395 women were randomized to the Healthy Options intervention. The average number of psychosocial support group sessions attended by the women was 6 (range 0–11 sessions). At the first follow-up at 6 weeks postpartum, data were collected from 649 (87.5%) women, corresponding to a response rate of 85.6% in the control group and 89.1% in the intervention group. At the 9-month follow-up, data were collected from 641 (86.4%) women, corresponding to a response rate of 83% for the control group and 89.4% for the intervention group. Reasons for loss to follow-up included moved out of the study area (~43%), participant not reachable (~41%), death (~11%), and refusal (~5%).

There was a decrease in mean depression score (PHQ-9 score) for intervention and control arms at the different time points; 6 weeks follow-up and 9 months follow-up. Mean difference in depression score between intervention and control groups measured at 6 weeks post intervention follow-up was -3.56 (-4.55 - -2.56) and at 9 months post intervention follow-up was -1.03 (-1.86, -0.19), these results were statistically significant at $p<0.01$ and $p<0.05$ respectively. More details on the Healthy Options baseline and end-line health outcomes have been reported elsewhere [16, 17].

During the follow-up period, trial findings show 2 deaths in each arm at 6 weeks, and 4 deaths in Healthy Options arm and 6 deaths in control arm at 9 months. YLL for control arm is 76 at 6 weeks and 228 at 9 months. YLL for Healthy Options is 76 at 6 weeks and 152 at 9 months. YLD for control arm is 21.9 at 6 weeks and 6.7 at 9 months. YLD for Healthy Options is 9.1 at 6 weeks and 6.1 at 9 months. Healthy Options had 12.8 fewer DALYs at 6 months (85.1 DALYs compared to 97.9 DALYs) and 76.6 fewer DALYs at 9 months (158.1 DALYs compared 234.7 DALYs) compared to control (Table 1).

### Healthy Options intervention cost

Total inflation adjusted cost of Healthy Options intervention was approximately $319,729 for the three years: year 1 ($43,716), year 2 ($133,221) and year 3 ($142,792) (Table 2). Cost distribution was as follows: buildings/space 5.2% of the total cost; furniture 0.4%; human resources 83.3%; office consumables 1.2%; training peer facilitators, supervisors and health care providers involved printing, transport and time compensation was 1.7% of total cost; expenses related

**Table 1. Disability-adjusted life years for control and intervention.**

| | Cases | Duration (years) | Disability Weight | YLD | Deaths | Average age at death | Standard life expectancy at death | YLL | DALYs |
|---|---|---|---|---|---|---|---|---|---|
| | | | | EUDC | | | | | |
| **Baseline** | 317 | 0.5 | 0.396 | 62.766 | 0 | 30 | 38 | 0 | 62.766 |
| **6 Weeks** | 111 | 0.5 | 0.396 | 21.978 | 2 | 30 | 38 | 76 | 97.978 |
| **9 Months** | 34 | 0.5 | 0.396 | 6.732 | 6 | 30 | 38 | 228 | 234.732 |
| | | | | EUDC plus Healthy Options | | | | | |
| | Cases | Duration (years) | Disability Weight | YLD | Deaths | Average age at death | Standard life expectancy at death | YLL | DALYs |
| **Baseline** | 361 | 0.5 | 0.396 | 71.478 | 0 | 30 | 38 | 0 | 71.478 |
| **6 Weeks** | 46 | 0.5 | 0.396 | 9.108 | 2 | 30 | 38 | 76 | 85.108 |
| **9 Months** | 31 | 0.5 | 0.396 | 6.138 | 4 | 30 | 38 | 152 | 158.138 |

to psychosocial support group sessions, which included travel and time reimbursement and refreshments was 6.5% of the total cost (excluding personnel cost); and communication expenses accounted for 1.6% of total cost. Cost of the intervention per woman was $883.

## Healthy Options intervention cost-effectiveness

Cost of EUDC was $398 and included training cost for health care providers in control arm (Table 3). Incremental cost-effectiveness ratio of Healthy Options at 6 weeks postpartum is $89,699 per mean decrease in depression score (Table 3). Incremental cost-effectiveness ratio at 9 months postpartum is $310,030 per mean decrease in depression score. ICER per DALY averted is $24,754 at 6 months and $4,169 at 9 months (Table 3). With sensitivity analysis, ICER per DALY averted at 6 weeks postpartum ranged from $17,906 to $54,354 and from $3,353 to $4,977 at 9 months (Table 4). Therefore, our findings show that Healthy Options is not cost-effective because the ICER per DALY averted is above 2017 GDP per capita for Tanzania, $975.9 [31].

## Discussion

Depression is a common cause of disability in HIV infected pregnant women. Treatment for prenatal depression is beneficial for health and wellbeing of the woman and her child [32]. Healthy Options has been found to be effective in reducing depressive symptoms in perinatal

**Table 2. Inflation adjusted cost of Healthy Options implementation (in US$).**

| Inputs | Year 1 | Year 2 | Year 3 | Total | Proportion |
|---|---|---|---|---|---|
| Space | 5,883.6 | 5,474.5 | 5,330.4 | 16,688.5 | **5.2** |
| Furniture | 449.2 | 417.9 | 406.9 | 1,274.0 | **0.4** |
| Human resources | 27,238.8 | 114,848.1 | 124,399.9 | 266,486.8 | **83.3** |
| Office consumables | 3,912.0 | - | - | 3,912.0 | **1.2** |
| Training (peer facilitators and supervisors) | - | - | - | - | **-** |
| Printing | 116.9 | - | 169.1 | 286.0 | **0.1** |
| Transport | - | 1,099.2 | - | 1,099.2 | **0.3** |
| Stipend | - | - | 4,182.1 | 4,182.1 | **1.3** |
| Therapy sessions | - | - | - | - | **-** |
| Travel reimbursement | 4,457.0 | 4,147.1 | 4,037.9 | 12,642.1 | **4.0** |
| Refreshments | - | 5,269.2 | 2,763.1 | 8,032.3 | **2.5** |
| Communication | 1,658.5 | 1,965.0 | 1,502.5 | 5,125.9 | **1.6** |
| **Total** | **43,715.9** | **133,221.1** | **142,791.9** | **319,728.9** | **100** |

**Table 3. Cost-effectiveness of Healthy Options intervention (in US$).**

| | Outcome 1 | | | | | | | |
|---|---|---|---|---|---|---|---|---|
| | Depression score | | Cost of intervention | Incremental Cost | Incremental Outcome | | ICER | |
| | 6 Weeks mean (SD) | 9 Months mean (SD) | | | 6 Weeks | 9 Months | 6 Weeks | 9 Months |
| | | | | | mean-difference (95% CI) | mean-difference (95% CI) | | |
| EUDC | 6·9 (3·9) | 3·6 (3·4) | 398 | | | | | |
| EUDC plus Healthy Options | 3·4 (4·2) | 2.6 (3.7) | 319729 | 319331 | 3·56 (4·55, 2·56) | 1.03 (1.86, 0.19) | 89699 | 310030 |
| | Outcome 2 | | | | | | | |
| | DALYs | | Cost of intervention | Incremental Cost | Incremental DALYs (averted) | | ICER | |
| | 6 Weeks | 9 Months | | | 6 Weeks | 9 Months | 6 Weeks | 9 Months |
| EUDC | 98 | 234.7 | 398 | | | | | |
| EUDC plus Healthy Options | 85.1 | 158.1 | 319729 | 319331 | -12.9 | -76.6 | 24754 | 4169 |

women with HIV in Tanzania compared to EUDC, with these positive effects observed at 6 weeks and 9 months postpartum. The unit cost of Healthy Options is $883 per woman. ICER per DALY averted is $24,754 at 6 months and $4,169 at 9 months.

Unit cost of Healthy Options is within range reported in similar studies where intervention cost ranged from cost saving to £1200 (approximately $1540) per woman [33, 34]. Our total cost estimates are for 36 months, a relatively long period compared to similar studies that have reported cost for shorter timelines [30, 33]. Findings from the trial show that not all 395 women randomised into the Healthy Options intervention participated, only 340 received the intervention [17]. With the same level of resources expended in this intervention, more women could have been reached with depression intervention if all randomized women had participated. Reasons behind non-participation were likely not related to background characteristics explored in the baseline study, the two groups had similar characteristics [17]. Moreover, cost of access could not have been a deterrent to participation as the women were compensated for travel and time, equivalent to $5.5. Other possible explanatory factors could be depression itself, since being depressed is associated with low/poor utilization of needed health services. It is important to identify and address factors that discourage depression treatment utilization. Efficiency of health service provision relies on better use of available resources.

**Table 4. Sensitivity analysis of Healthy Options incremental cost-effectiveness ratio (in US$).**

| | 6 weeks postpartum | 9 months postpartum |
|---|---|---|
| **Sensitivity analysis- high values** | | |
| Incremental Cost of Healthy Options (US$) | 383,197.4 | 383,197.4 |
| Incremental Effectiveness of Healthy Options (DALYs averted) | 21.4 | 77.0 |
| Incremental Cost-effectiveness Ratio of Healthy Options (US$) | 17,906.4 | 4,976.5 |
| **Sensitivity analysis- low values** | | |
| Incremental Cost of Healthy Options (US$) | 255,465.0 | 255,465.0 |
| Incremental Effectiveness of Healthy Options (DALYs averted) | 4.7 | 76.2 |
| Incremental Cost-effectiveness Ratio of Healthy Options (US$) | 54,354.3 | 3,352.5 |

Our study worked with lay CBHWs who were trained to provide treatment to women and the results demonstrated effectiveness of the CBHW model. Tanzania faces the challenge of scarcity of mental health human resources and the CBHW model may offer a feasible alternative to the expense of employing professional mental health care providers. However, since CBHWs do not have a formal education in provision of mental health care, thorough training and supportive supervision are critical. With adequate support, non-specialist mental health providers can effectively provide basic services for depression and referral for more complex cases [18].

Results from the intervention study demonstrated a sustained reduction in depressive symptoms among perinatal women with HIV who participated in the Healthy Options intervention compared with the control group. Even at 9 months postpartum, there was a significant difference in severity of depressive symptoms between the intervention participants and controls. Nonetheless, Healthy Options was not cost-effective. Our cost-effectiveness analysis has not captured all the benefits associated with the Healthy Options intervention. First, we did not include improved secondary benefits, such as improved self-efficacy, HIV-related stigma, household food security and appreciative relationship with partner. Inclusion of secondary benefits may have improved cost-effectiveness of Healthy Options intervention. Second, we did not consider societal perspectives and long-term outcomes of the intervention; these considerations may have improved cost-effectiveness of Healthy Options.

Identification and treatment of depression during pregnancy and postpartum have been found to be effective and cost-effective in similar studies [30, 35] with short and long-term benefits to both the woman and her child/children. Benefits include improved mother–infant interaction, better nutrition, improved cognitive development and growth, reduced diarrhoeal episodes, increased immunization rates, ART adherence, better HIV prognosis and survival for the child and mother [36]. The day-to-day interaction between infants and their mothers impacts neurological, cognitive, emotional, and social development throughout childhood with an impact on adult health and socio-economic wellbeing.

Long run benefits of treating perinatal depression comprise improved health and saved resources. For example, perinatal depression has been found to cause 3.2 million cases of stunting in children and an economic cost of $14.5 billion in LMICs [2]. The economic cost consists of future revenue lost by adults because of exposure to psychosocial risks early in life. Exposure to psychosocial risks early in life is linked to poor long-term health and well-being of the child. In addition, this exposure can result in poor child nutrition, child stunting, poor school attainment and negative impact on wages in adulthood.

Mental health treatment has its greatest impact outside the health sector, so it becomes complex to capture mental health intervention benefits comprehensively. For example, long-term emotional, behavioural and cognitive damage to the child has more effect on employment, hence more cost-savings can be realized in terms of reduced unemployment-related cost than reduced health care cost if depression is treated in this population [18]. Maternal depression is a chronic condition with long-term economic consequences, as a result full benefit of treatment may not be captured in the short-term as it may take some years for benefits to be realized.

## Study limitations

Results from this study are applicable to our study sites and cannot be generalised to other contexts. The Healthy Options study was a cluster randomised clinical trial and its participants were specifically selected following inclusion criteria that may not be representative of real community contexts. Also, conditions in a clinical trial may not reflect real-world behaviours

of both health care providers and clients, which can be characterised by higher drop-out rates, lower utilization, and reduced adherence to treatments.

Another limitation is that our study has not shown the long-term cost-effectiveness of treating depression in HIV positive perinatal women. As indicated above, treatment of depression in women may have long-term benefits for the mother and her child/children. And would possibly provide evidence of wider cost-savings from depression treatment and thus improve cost-effectiveness of the Healthy Options intervention.

Moreover, this study allowed for clustering in incremental outcome but not in costs. Sample size calculations incorporated clustering anticipated in outcomes but not in costs. According to Gomes et al. [37], use of methods that ignore clustering underestimate statistical uncertainty and can lead to inaccurate point estimates. Nonetheless, this study fills a critical knowledge gap by providing valuable information on the cost and cost-effectiveness of treating perinatal depression among women living with HIV accessing PST and CBT in Tanzania.

## Conclusions

Depression is a common cause of disability in HIV infected pregnant women in Tanzania. Results from this study show that Healthy Options, a psychosocial support group intervention, is effective but not cost-effective in reducing depressive symptoms among pregnant women with HIV in Tanzania. Healthy Options worked with CBHWs to successfully deliver mental health services; this likely represents a more feasible model of depression management given the scarcity of mental health specialists in Tanzania.

## Supporting information

**S1 Table. Resource inputs and costs (in US$).**
(DOCX)

## Acknowledgments

We would like to express our sincere gratitude to the participants and staff at the study reproductive and child health centres that made this study possible.

## Author Contributions

**Conceptualization:** Happiness Pius Saronga, Sylvia Kaaya, Mary C. Smith Fawzi.

**Data curation:** Happiness Pius Saronga, Sylvia Kaaya.

**Formal analysis:** Happiness Pius Saronga.

**Funding acquisition:** Sylvia Kaaya, Mary C. Smith Fawzi.

**Investigation:** Happiness Pius Saronga, Mary C. Smith Fawzi.

**Methodology:** Happiness Pius Saronga, Sylvia Kaaya, Mary C. Smith Fawzi.

**Project administration:** Mary C. Smith Fawzi.

**Resources:** Sylvia Kaaya, Mary C. Smith Fawzi.

**Software:** Happiness Pius Saronga.

**Supervision:** Sylvia Kaaya, Mary C. Smith Fawzi.

**Validation:** Happiness Pius Saronga, Sylvia Kaaya, Mary C. Smith Fawzi.

**Visualization:** Happiness Pius Saronga.

**Writing – original draft:** Happiness Pius Saronga, Sylvia Kaaya, Mary C. Smith Fawzi.

**Writing – review & editing:** Happiness Pius Saronga, Sylvia Kaaya, Mary C. Smith Fawzi.

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
