## [Decision Letter · Decision Letter 0]

30 Jul 2024

PMEN-D-24-00182

Cost-effectiveness of the Healthy Options group psychosocial intervention for perinatal women living with HIV and depression in Tanzania

PLOS Mental Health

Dear Dr. Saronga,

Thank you for submitting your manuscript to PLOS Mental Health. After careful consideration, we feel that it has merit but does not fully meet PLOS Mental Health’s publication criteria as it currently stands. Therefore, we invite you to submit a revised version of the manuscript that addresses the points raised during the review process.

We look forward to receiving your revised manuscript.

Kind regards,

Abigail Mae Hatcher, PhD

Academic Editor

PLOS Mental Health

Journal Requirements:

1. We note that your Data Availability Statement is currently as follows: The dataset supporting the conclusions of this article is included within the article.

Additional Editor Comments (if provided):

Kindly address the comments raised by reviewers.

Reviewers' comments:

Reviewer's Responses to Questions

**Comments to the Author**

1. Does this manuscript meet PLOS Mental Health’s publication criteria? Is the manuscript technically sound, and do the data support the conclusions? The manuscript must describe methodologically and ethically rigorous research with conclusions that are appropriately drawn based on the data presented.

Reviewer #1: Yes

Reviewer #2: Yes

2. Has the statistical analysis been performed appropriately and rigorously?

Reviewer #1: Yes

Reviewer #2: Yes

3. Have the authors made all data underlying the findings in their manuscript fully available (please refer to the Data Availability Statement at the start of the manuscript PDF file)?

Reviewer #1: Yes

Reviewer #2: No

4. Is the manuscript presented in an intelligible fashion and written in standard English?

Reviewer #1: Yes

Reviewer #2: Yes

5. Review Comments to the Author

Reviewer #1: RE: PMEN-D-24-00182

Cost-effectiveness of the Healthy Options group psychosocial intervention for perinatal women living with HIV and depression in Tanzania

PLOS Mental Health

SUMMARY: This paper reported the clinical benefit and economic assessment of the Healthy Options (intervention title) intervention from a cluster randomized clinical trial to address mental health issues for perinatal women living with HIV in Tanzania. All participants received enhanced usual care for depression treatment. Intervention group received, in extra, problem-solving therapy to promote awareness of one’s own strengths and assertive communication skills, acknowledging problems and identifying strategies for addressing them, focusing on positive thinking approaches and using dreaming to develop action plans and foster hope. Cognitive behavioral therapy was additionally provided to only those with elevated depression in the intervention group. The trial showed an improvement in the clinical outcomes—survival and depression. Standard methods were used and reproducible. Among those, the usage of PHQ-9 as a common depression measure helps for future literature synthesis for multiple purposes. It is much appreciated to have a “negative cost-effectiveness” finding reported since the economic evaluation literature has had an inclination on reporting “positive cost-effectiveness” finding, aka a publication bias. Healthy Options was not cost-effective compared to usual care under the standard willingness to pay threshold of up to 3 times of the country’s GDP per capita.

MAJOR ISSUES

• Main goal of the paper: a) As the title holds on cost-effectiveness, much of report on study outcomes are deemed redundant. Should only study outcomes on death and depression be reported because these are key drivers to the estimate of the denominator of the incremental cost-effectiveness ration. b) The paper is informational though. If no other paper reports all study outcomes, this paper may use an alternate title to include “Clinical and economic benefits of Healthy Options”...

• There are too many tables. Consider deleting redundant and combining them. Table 1 can be in Supplemental. Table 2 can be included in the base-case analysis reporting for the cost-effectiveness. Table 3. Delete. Table 5: Delete. Table 7 and 8 may be combined (Also, I don’t think there was a need to report unadjusted costs. I would suggest proceeding with the reporting with the adjusted number and holding as is the methods discussing the unadjusted and adjusted costs).

• Table 9: Please, always use “incremental” cost-effectiveness ratio. Economic evaluation is an incremental analysis. Better wording in the first column is needed if readers would like to skim through the tables first. Cost -> Implementation cost or Costs of Healthy Options (implementation). It is confusing to use a negative DALY in this table. Should this DALY number be positive since the intervention group had better survival and lower depression scores?

• The authors separated intervention vs. non-intervention costs. However, how could you differentiate them? Resources used were recorded; but how did the recording help you arrive at the estimators such as 75%?

MINOR ISSUES

• Line 392: You can quickly and roughly convert £1200 into USD2017 for readers’ ease

• Use consistent wording across tables

Reviewer #2: In consideration of no.3 (above) and Table 1 of the manuscript please indicate the amount instead of labeling it as "lump sum" for the sake of clarity and replication. Otherwise the manuscript is well written.

6. PLOS authors have the option to publish the peer review history of their article (what does this mean?). If published, this will include your full peer review and any attached files.

**Do you want your identity to be public for this peer review?** For information about this choice, including consent withdrawal, please see our Privacy Policy.

Reviewer #1: No

Reviewer #2: **Yes: **Mah Wasi Asombang

---

## [Editor Report · Decision Letter 1]

17 Sep 2024

PMEN-D-24-00182R1

Cost-effectiveness of the Healthy Options group psychosocial intervention for perinatal women living with HIV and depression in Tanzania

PLOS Mental Health

Dear Dr. Saronga,

Thank you for submitting your manuscript to PLOS Mental Health. After careful consideration, we feel that it has merit but does not fully meet PLOS Mental Health’s publication criteria as it currently stands. Therefore, we invite you to submit a revised version of the manuscript that addresses the points raised during the review process.

We look forward to receiving your revised manuscript.

Kind regards,

Abigail Mae Hatcher, PhD

Academic Editor

PLOS Mental Health

Journal Requirements:

Additional Editor Comments (if provided):

The responses to reviewers are strong and this paper is ready for publication. On a minor copy-edit level, may I kindly suggest you spell out the intervention name rather than use the acronym "HO" throughout?
---

## [Editor Report · Decision Letter 2]

29 Oct 2024

Cost-effectiveness of the Healthy Options group psychosocial intervention for perinatal women living with HIV and depression in Tanzania

PMEN-D-24-00182R2

Dear Dr. Saronga,

We are pleased to inform you that your manuscript 'Cost-effectiveness of the Healthy Options group psychosocial intervention for perinatal women living with HIV and depression in Tanzania' has been provisionally accepted for publication in PLOS Mental Health.

Best regards,

Abigail Mae Hatcher, PhD

Academic Editor

PLOS Mental Health